# Prolonged Supplementation of Ozonated Sunflower Oil Bestows an Antiaging Effect, Improves Blood Lipid Profile and Spinal Deformities, and Protects Vital Organs of Zebrafish (*Danio rerio*) against Age-Related Degeneration: Two-Years Consumption Study

**DOI:** 10.3390/antiox13010123

**Published:** 2024-01-19

**Authors:** Kyung-Hyun Cho, Ashutosh Bahuguna, Dae-Jin Kang, Ji-Eun Kim

**Affiliations:** Raydel Research Institute, Medical Innovation Complex, Daegu 41061, Republic of Korea; ashubahuguna@raydel.co.kr (A.B.); daejin@raydel.co.kr (D.-J.K.); ths01035@raydel.co.kr (J.-E.K.)

**Keywords:** ozonated sunflower oil, antioxidant, dyslipidemia, kidney, liver, ovary, radio imaging, senescence, testis, zebrafish

## Abstract

Ozonated sunflower oil (OSO) is renowned for its diverse therapeutic benefits. Nonetheless, the consequences of extended dietary intake of OSO have yet to be thoroughly investigated. Herein, the effect of 2-year dietary supplementation of OSO was examined on the survivability, obesity, skeletal deformities, swimming behavior, and liver, kidney, ovary, and testis function of zebrafish. Results showed that the zebrafish feed supplemented with 20% (*wt*/*wt*) OSO for 2 years emerged with higher survivability and body weight management compared to sunflower oil (SO) and normal diet (ND)-supplemented zebrafish. Radio imaging (X-ray)-based analysis revealed 2.6° and 15.2° lower spinal curvature in the OSO-supplemented groups than in the SO and ND-supplemented groups; consistently, OSO-supplemented zebrafish showed better swimming behavior. The histology analysis of the liver revealed the least fatty liver change and interleukin (IL)-6 generation in the OSO-supplemented group. Additionally, a significantly lower level of reactive oxygen species (ROS), apoptotic, and senescent cells were observed in the liver of the OSO-supplemented zebrafish. Also, no adverse effect on the kidney, testis, and ovary morphology was detected during 2 years of OSO consumption. Moreover, lower senescence with diminished ROS and apoptosis was noticed in the kidney and ovary in response to OSO consumption. The OSO supplementation was found to be effective in countering age-associated dyslipidemia by alleviating total cholesterol (TC), triglycerides (TG), low-density lipoproteins (LDL-C) and elevating high-density lipoproteins (HDL-C)/TC levels. Conclusively, prolonged OSO consumption showed no adverse effect on the morphology and functionality of vital organs; in fact, OSO supplementation displayed a protective effect against age-associated detrimental effects on spinal deformities, vital organ functionality, cell senescence, and the survivability of zebrafish.

## 1. Introduction

Sunflower oil (SO) extracted from sunflower seeds is among the major oils used for culinary purposes and has a substantial nutritional and health effect. Linolenic acid and oleic acids are the two primary unsaturated fatty acids that account for 90% of SO’s total higher fatty acid content [1]. Additionally, palmitic acid and steric acid have a substantial aggregate and make a significant chunk of unsaturated fatty acids of SO [1,2]. In addition to fatty acids, SO is a valuable resource of dietary vitamin E (50–150 mg/100 g), primarily α-tocopherol (90%) [3], and minerals like manganese [4]. Although the phytoconstituent of SO is the inherited behavior of the seeds used to extract the SO, it can change substantially with a variety of sunflowers and environmental factors [2]. Many efforts have been made to improve the constituents of sunflower seeds to enhance the nutritional value and bifunctionality of SO [1]. Mostly, these efforts are focused on the genetic level to obtain sunflower seeds with the desired quality [1].

Ozonation is an additional way that substantially affects the biofunctionality of SO. In ozonation, ozone is pumped into oil where ozone reacts with unsaturated fatty acids via electrophilic addition, leading to the formation of several products such as hydroperoxides and criegee ozonide that impart diverse biofunctionality to the oil [5]. Notably, during ozonation, only unsaturated sites are affected without any impact on the oil’s saturated fatty acids and other phytonutrients [6]. Various vegetable oils have been successfully used to develop stable ozonated oils that have been extensively used for antimicrobial agents for topical application [7].

Among the different ozonated oils, SO has been greatly acknowledged for its antimicrobial [8], anti-inflammatory, antioxidant [8,9], and wound-healing role [10,11]. As an antioxidant, ozonated SO (OSO) proved efficient in upregulating cellular antioxidants (superoxide dismutase) and reduced thiobarbituric acid reactive substances (TBARS) in the gastric mucosa of rats [12]. Also, the role of OSO in enhancing glutathione peroxidase (GSH-PX) and, consequently, its impact on glutathione (GSH) metabolism has been reported [12]. In one of the recent studies, we interpreted the positive impact of OSO on the structural modification of high-density lipoprotein (HDL) and HDL-associated antioxidant enzyme paraoxonase (PON)-1 [8]. Furthermore, the effective contribution of OSO in averting hyperlipidemia induced by a high-cholesterol diet [9], along with its role as a modulator of inflammation and influence on fibroblast proliferation, collagen fiber remodeling, and tissue regenerative activity, has been noted [12]. Numerous studies have elucidated the crucial therapeutic benefits of OSO in combating various ailments [13,14,15,16]; nevertheless, investigations on the prolonged dietary intake of OSO and its associated health consequences have yet to be initiated. Therefore, this research was initiated to assess the impact of sustained long-term consumption of OSO on the health of zebrafish, aiming to explore the potential of OSO as a functional food.

Taking this into account, the present study was performed to examine the impact of OSO supplementation in the diet for 2 years on the survivability, body weight, structural deformities, swimming behavior, lipid profile, and functionality of the zebrafish’s vital organs (liver, kidney, testis, and ovary). Furthermore, an assessment of the effect of OSO on age-related cellular senescence was conducted.

## 2. Materials and Methods

### 2.1. Materials

The ozonated sunflower oil (Raydel^®^, Bodyone, Flambo oil, Thornleigh, NSW, Australia) with a characteristic Oleozone^®^ [17] was obtained from Rainbow and Nature Pty, Ltd. (Thornleigh, NSW, Australia). The sunflower oil (Ondoliva oil, Urzante, Tudela, Spain) was procured from a local supermarket in Daegu, Republic of Korea. Acridine orange (Cat#A9231), dihydroethidium (Cat#37291), 2 phenoxyethanol (Cat#P1126), 5-bromo-4-chloro-3-indolyl β D-galactopyranoside (X-gal, Cat#B4252), and oil red O (Cat#O0625) were purchased from Sigma-Aldrich (St. Louis, MO, USA). Unless specified, all remaining reagents were of high purity and utilized as provided.

### 2.2. Preparation of Sunflower Oil (SO) and Ozonated Sunflower Oil (OSO) Supplemented Diet

The normal tetrabit (ND, Tetrabit Gmbh D49304, 47.5% crude protein, 6.5% crude fat, 2.0% crude fiber, 10.5% crude ash, containing vitamin A (29,770 IU/kg), vitamin D3 (1860 IU/kg), vitamin E (200 mg/kg), and vitamin C (137 mg/kg); Melle, Germany), a regular zebrafish food, was blended with 20% SO (*wt*/*wt*) or 20% OSO (*wt*/*wt*). The 50 g SO or OSO (10 g × 5 times) underwent intermittent mixing with 200 g of ND. Following the addition of each portion of SO or OSO (10 g), thorough mixing was carried out to achieve the proper distribution of SO and OSO. The UV-Vis spectrophotometer was utilized to assess the quantification of SO and OSO absorbed/adsorbed onto ND. The SO and OSO attached to the ND were extracted by ethanol following the procedure outlined in the Supplementary Methods. The quantification of SO and OSO extracted from the ND was carried out using a UV spectrophotometer, utilizing the standard curve of SO and OSO within the concentration range of 0.5–2 mg/mL (Appendix A). The results indicated a recovery range of 73.8% to 89.1% for SO, corresponding to 14.7% to 17.8% of SO absorbed/absorbed in ND. Similarly, there was a recovery range of 75.3% to 94% for OSO, which is equivalent to 15% to 18.8% of OSO absorbed/absorbed in ND.

### 2.3. Zebrafish Aquaculture

Zebrafish were upheld in a water tank sustained at 28 °C temperature with constant automated aeration (Bioengineering Co., Daejeon, Republic of Korea) following the prescribed rules of care and use of laboratory animals [18,19] adopted by the Animal Care Committee and the Use of Raydel Research Institute (approval code RRI-20-003, approval date 3 January 2020). A light (14 h) and dark (10 h) photoperiod was maintained during the zebrafish husbandry. Zebrafish were fed with normal tetrabit.

### 2.4. Zebrafish Fed with Sunflower Oil (SO) and Ozonated Sunflower Oil (OSO)

The 8-week-aged zebrafish (n = 240) underwent random allocation to three cohorts (n = 80, each group). Within each cohort (n = 80), the zebrafish were further divided into 4 sets, each comprising 20 individuals (n = 20). Zebrafish in cohort I received normal tetrabit (ND and considered as control), while the zebrafish in cohort II were maintained in ND supplemented with 20% SO (*wt*/*wt*). Likewise, the zebrafish in cohort III were maintained in ND supplemented with a 20% OSO (*wt*/*wt*) diet. All the cohorts were maintained under similar environmental conditions. Zebrafish survivability in each cohort was recorded at different time points (0 to 24 months).

Correspondingly, the zebrafish’s weight across all three cohorts was determined gravimetrically at 0, 12, 18, 21 and 24 months after anesthetization (using 0.1% solution of 2-phenoxyethanol). At the beginning (0 months), the body weight of all the zebrafish (n = 80) from each cohort was assessed by submerging them for 2 min in 0.1% solution of 2-phenoxyethanol. The anaesthetized zebrafish were surface-dried with tissue paper, and their weight was determined using an electronic weighing balance (Ohasus, Parsippany-Troy Hills, NJ, USA). For the further (12, 18, 21 and 24 months) body weight analysis, one set of the zebrafish (n = 20) from each cohort was analyzed as the earlier described method. Notably, the same set of zebrafish from each cohort was used for the weight analysis at 12, 18, 21 and 24 months.

Moreover, the swimming behavior of zebrafish within the three different groups was assessed by observing tail fin moments, estimating swimming speed, and documenting swimming patterns through digital recording. The “Tracker video analysis and modelling tool (available at https://physlets.org/tracker accessed on 16 September 2023)” was utilized to ascertain the swimming trajectory and quantify the swimming speed of zebrafish.

### 2.5. Radiology Imaging

Zebrafish (28 months old) from the different groups were anesthetized by submerging them into a 0.1% solution of 2-phenoxyethanol. The anesthetized zebrafish were processed for whole skeleton radiology imaging using an X-ray machine (Woorien, Myvet dent model, Hwaseong, Republic of Korea) at an accelerating voltage of 50 kV with 2.5 mA and 0.3 s f stop (Siji W Animal Medical Center, Daegu, Republic of Korea). The obtained image was processed by IntoVetGE, imaging software version 1.5.14.2 (IntoCNS, Seoul, Republic of Korea). For the quantification of the spinal bending, an angle was determined through Image J software (http://rsb.info.nih.gov/ij/, 1.53 version assessed on 30 January 2023). The angle of spinal bending was calculated between vertebra 1 and 24, considering vertebra 5 a central point.

### 2.6. Histology and Immunohistochemical Analysis

At 24 months, zebrafish (n = 10) from different groups were sacrificed using the hypothermic shock [9], and different organs (liver, kidney, ovary, and testis) were surgically removed and preserved in 10% formalin. The tissue (n = 5 for each group) from the different organs was individually dehydrated using ethanol and subsequently embedded into paraffin wax, which was followed by tissue sectioning (7 μm thick). Liver, kidney, ovary, and testis tissue sections underwent the processing for hematoxylin and eosin (H&E) staining [20] to assess morphological changes.

The tissue section from the liver was also processed for the fatty liver changes using oil red O staining, following the method described previously [21]. Briefly, oil red O stain (final, 3 mg/mL) was spread over the tissue section. After 10 min incubation, the stained area was washed with tap water and observed under the microscope (Motic SMZ 168; Hong Kong).

Immunohistochemical (IHC) staining was employed for interleukin 6 (IL-6) production in hepatic tissue as a previously described method [22]. The hepatic tissue section (7 μm thick) was immersed with IL-6 specific primary antibody (200 times diluted; (ab9324, Abcam, London, UK) and incubated at 4 °C for 18 h. After that, the tissue section was developed by using horseradish peroxidase (HRP)-conjugated anti-IL-6 antibody (1000 time diluted) using EnVison+ System HRP labeled polymer kit (Code K4001, Dako, Glostrup, Denmark) and observed under a microscope (Motic SMZ 168; Hong Kong). To enhance the clarity of presentation, the IL-6-stained area (originally brown) was replaced with a red color when the brown color threshold value was set between 20 and 120 using Image J software.

### 2.7. Imaging for Reactive Oxygen Species (ROS) Production and Apoptosis Extent

The method described earlier [23] was employed to assess the production of reactive oxygen species (ROS) in the tissue section (n = 5 for each group) using dihydroethidium (DHE) fluorescent staining. Briefly, a 250 μL of DHE solution (final, 30 μM) was added to the tissue section. Following 30 min incubation in the dark at room temperature (RT), the stained area was rinsed with tap water and observed using a fluorescent microscope (Nikon Eclipse TE2000, Tokyo, Japan) with excitation at 588 nm and emission at 618 nm wavelength. The extent of apoptosis was examined using acridine orange (AO) fluorescent staining, following a method described previously [24] with slight modification. In summary, a tissue section (n = 5 for each group) was treated with 250 μL of AO (5 μg/mL). After 30 min incubation in the dark at RT, the stained section was washed two times with tap water and subsequently visualized under a fluorescent microscope (Nikon Eclipse TE2000, Tokyo, Japan) at an excitation wavelength of 505 nm and an emission wavelength of 535 nm.

### 2.8. Senescence-Tied β Galactosidase Staining

A previously described senescence-associated β galactosidase (SA-β-gal) staining method [25] with slight modification was implemented to examine the senescent cells in the liver, kidney, testis, and ovary. Briefly, a tissue section (n = 5 for each group; 7 μm thick) underwent fixation using 4% paraformaldehyde and incubated for 5 min at room temperature (RT), which was followed by two washes with phosphate-buffered saline (PBS). The tissue section was flooded with X-gal solution (X-gal (final, 1 mg/mL) dissolved in citrate-phosphate buffer (pH 5.9) containing 5 mM each of ferric and ferro cyanide, 150 mM NaCl, 2 mM of MgCl_2_ and incubated in the dark for 16 h, following two times washing with PBS and visualization under the microscope (Motic SMZ 168; Hong Kong).

### 2.9. Analysis of Blood Lipid Profile and Biomarkers Associated with Liver Function

A 2 μL of blood was collected from the zebrafish’s heart and instantly mixed with 5 μL of PBS–ethylenediaminetetraacetic acid (EDTA, final 1 mM). The collected plasma proceeded for the quantification of total cholesterol (TC), triglycerides (TG), and high-density lipoproteins (HDL-C) using the commercial diagnostic kit (cholesterol, T-CHO, and TGs, Cleantech TS-S; Walko Pure Chemical, Osaka, Japan). In brief, 5 μL of serum was mixed with 200 μL of TC-specific reaction mixture (provided with the diagnostic kit). After 10 min incubation at 37 °C, the developed red color product corresponding to TG was quantified by taking absorbance at 490 nm using a microplate reader (BioRad iMark™; BioRad, Hercules, CA, USA). Similarly, 5 μL of serum was blended with 200 μL of TC-specific reaction mixture (provided with the diagnostic kit). After 10 min incubation at 37 °C, the colored product (corresponding to TG) was quantified by taking absorbance at 550 nm. For high-density lipoproteins (HDL-C) analysis, an equal amount of serum was amalgamated with separation solution (provided with the diagnostic kit) followed by 10 min centrifugation at 3000 rpm, and the 20 μL supernatant was mixed with reaction buffer (supplied with diagnostic kit). HDL-C was quantified by taking absorbance at 490 nm after 10 min incubation at 37 °C. The low-density lipoproteins (LDL-C) concentration in the blood plasm was calculated using the Friedwald equation: [TC–HDL–(TG/5)].

The commercial diagnostic kit (Asan Pharmaceutical, Hwasung, Republic of Korea) was used to quantify aspartate transaminase (AST) and alanine transaminase (ALT) levels in the serum, following the instructions suggested by the manufacturers. Briefly, 5 μL of serum was combined with 250 μL of either AST or ALT-specific solution, as supplied in the diagnostic kit. Following a 30 min incubation for AST or 60 min incubation of ALT at 37 °C, the mixture was then blended with 250 μL of the respective coloring reagent (AST or ALT-specific, provided in the diagnostic kit). After a subsequent 20 min incubation at RT, 250 μL of 0.4 N NaOH was introduced to halt the reaction. Finally, the AST and ALT were quantified by measuring absorbance at 490 nm.

### 2.10. Statistical Analysis

The results are depicted as the mean ± standard deviation (SD) from three distinct experiments. Statistical analysis was conducted using the Statistical Package for the Social Science (SPSS version 29.0; Chicago, IL, USA), employing one-way analysis or two-way analysis of variance (ANOVA) followed by Tukey’s multiple range test to identify significant distinctions among the groups. Prior to performing a parametric one-way ANOVA, the distribution’s normality was assessed through the Kolmogorov–Smirnov test.

## 3. Results

### 3.1. *Zebrafish Survivability and Body Weight*

The survivability of zebrafish in all the groups progressively decreases with the progression of time (Figure 1A). At the end of 12 months, supplementation of SO and OSO groups showed a nearly similar survivability of 96.3% and 95.4%, respectively, which is slightly better than the survivability observed in the ND group (92.5%). However, at the end of 17 months, the least survivability (87.5%) was observed in the SO group, which was followed by the ND group (90.0%). In contrast, the OSO feed group showed much higher survivability (93.9%). Consistent with 17 months, no changes in the zebrafish survivability were observed at 18 months of feeding in ND and OSO groups; unlike this, survivability decreased and reached 85.0% in the SO group. Interestingly, at 21 months onwards, OSO-supplemented groups displayed nearly constant survivability. Contrary to this, the ND and SO-supplemented groups revealed a sharp decline in survivability during 21 months to 24 months of feeding. At the final 24 months, 81.8%, 72.5%, and 71.2% survivability were observed in OSO, SO, and ND groups, i.e., 18.2%, 27.5%, and 28.8% lower than the survivability observed at the initial day. Results clearly demonstrated that OSO substantially affects zebrafish survivability, i.e., 9.3% and 10.6% better than the zebrafish survivability observed in SO and ND groups after 24 months of feeding.

A time-dependent change in the body weight of zebrafish was recorded up to 24 months, and results are depicted in Figure 1B,C. The body weight change in zebrafish was significantly influenced (*p* < 0.001) by both time and type of food (ND, SO, and OSO supplementation). After 12 months, the consumption of SO-supplemented group body weight (520 ± 21 mg) increased by 72.2% compared with its initial day value (301 ± 11 mg). Contrary to this, 30.4% and 29.3% changes in body weight were observed after 12 months in OSO (392 ± 16 mg) and ND (389 ± 13 mg) fed groups conversely to their initial day weight. The maximum bodyweight (886 ± 58 mg) was detected in the SO-fed group, i.e., 2.0-fold (439 ± 20 mg) and 1.6-fold (541 ± 30 mg) fold higher than the bodyweight of OSO and ND-fed groups at 18 months. From 18 months onwards, the body weight of OSO and ND-supplemented groups remained almost constant. Contrary to this, a sharp decline in body weight was perceived in the SO-supplemented group at 24 months than at 18 months. At 24 months, a significantly 1.3-fold (*p <* 0.001) and 1.2-fold (*p <* 0.001) higher body weight was noticed in the SO-supplemented group than in the body weight of OSO and ND-supplemented groups. The results imply that OSO supplementation did not affect the body weight, unlike SO supplementation, which severely enhances the increase in body weight.

### 3.2. Radiology Imaging for the Structural Deformities and Swimming Behavior of Zebrafish

Age-associated structural (backbone) bending was determined at 28 months post-feeding in ND, SO, and OSO-supplemented groups, and the results were compared with ND feed in 14-month-old young zebrafish (as a reference). The visual examination revealed no backbone structural deformities in ND and OSO-supplemented groups (Figure 2A). Unlike this, severe backbone bending was observed in the SO-supplemented group (Figure 2A). Further bending of the zebrafish backbone in different groups was quantified by radio imaging of the spine (Figure 2B–D and Appendix A). Radio imaging revealed a 164.1 ± 2.1° spinal angle in the ND feed of 14-month-old zebrafish, which is used as a reference to evaluate the alteration in the spinal bending in ND, SO, and OSO-supplemented groups. In the ND-supplemented group, a 152.1 ± 2.8° of spinal bending was observed, i.e., significantly 7.2% (*p* < 0.005) more than the reference spinal bending of 14-month-old zebrafish. Contrary to the ND group, the OSO group showed non-significant changes in spinal bending (156.3 ± 1.6°) compared to the reference (14-month-old zebrafish). While compared to ND, the OSO group exhibited 2.6° retardation in spinal bending. The most adverse results were observed in the SO-supplemented group where severe spinal bending was noticed, i.e., significantly 19.1% (*p* < 0.001), 12.2% (*p* < 0.001), and 15.2% (*p* < 0.001) more altered than the 14-month-old, ND, and OSO-supplemented groups.

Consistent with the findings of spinal bending, a difference in swimming behavior was observed between the groups (Appendix A). The OSO-supplemented group exhibited a significantly 1.9-fold (*p* < 0.012) and 2.1-fold (*p* < 0.009) higher swimming speed as compared to ND and SO-supplemented groups, respectively. The OSO-supplemented group exhibited a robust swimming pattern and achieved the highest average swimming speed of 14.8 ± 1.5 cm/s, surpassing the ND (7.8 ± 1.7 cm/s) and SO (7.3 ± 2.7 cm/s)-supplemented zebrafish swimming speeds (Figure 3 and Appendix A).

A collective outcome revealed the impact of prolonged OSO consumption on preventing age-associated spinal deformities, tissue regeneration, and, consequently, their swimming and locomotory behavior.

### 3.3. Evaluation of the Liver Section

The H&E staining results, as depicted in Figure 4A,B, revealed a dilatated portal vein with visible hepatic degeneration in the SO-supplemented group. In contrast, no hepatic degeneration was noticed in the OSO-supplemented groups; however, a minor hepatic degeneration was noticed around the portal vein of the ND-supplemented group. Results also exhibited the presence of neutrophils in the vicinity of the portal vein across the groups. However, the maximum amount was observed in ND, which was followed by SO and OSO-supplemented groups that account for 21.5%, 18.0%, and 15.9% of the H&E-stained area, respectively. Also, the presence of lipid accumulation (pointed by the yellow arrow) was observed only in the SO-supplemented group, signifying the impact of high amount and prolonged consumption of SO on liver steatosis. The result suggests no adverse effect of OSO supplementation over a long time on hepatic tissue; even more, OSO displayed a positive impact on age-associated adversity, which was evidenced by a 1.3-fold (*p* < 0.045) reduced H&E-stained area compared to the ND-supplemented group.

Cell senescence is an ideal marker for aging and age-related disorders. The SA-β-gal activity in the hepatic tissue of the ND, SO, and OSO-supplemented groups was evaluated, and results are depicted in Figure 4A,B. Age-related cell senescence, which was evident by a blue-stained area (denoted by a red arrow), was observed in the ND-supplemented group. In contrast, the supplementation of SO for 24 months significantly (*p* < 0.032) elevated hepatocyte senescence. Contrary to this, OSO supplementation displayed a significant 2.3-fold (*p* < 0.041)-reduced senescent stained area compared to SO-supplemented groups, signifying the importance of OSO in preventing the liver from age-related adversity.

The ROS level in the hepatic tissue at 24 months of ND, SO, and OSO supplementation was evaluated by DHE fluorescent staining. As depicted in Figure 4A,B, the highest ROS production was noticed in the SO group, which was significantly 1.6-fold (*p* < 0.001) more than the ROS level detected in the ND group, thus indicating the impact of long-term consumption of SO on hepatic ROS production. The least ROS production was perceived in the OSO-supplemented group, which was significantly 4.1 (*p* = 0.001) and 7.1-fold (*p* < 0.001) lower than the ROS production detected in ND and SO supplement groups, respectively. The results clearly indicated OSO’s influential role in protecting the liver from age-induced ROS generation and oxidative stress.

The extent of apoptosis evaluated by AO fluorescent staining revealed the highest apoptosis in the SO-supplemented group (Figure 4A,B). The ND and OSO-supplemented groups quantified approximately 1.5-fold (*p* < 0.001) reduced apoptosis compared to SO. Notably, no conspicuous variance in hepatic apoptosis was observed between ND and OSO-supplemented groups, which signifies that prolonged consumption of OSO did not affect the induction of apoptosis in the liver, asserting it as safe to consume. The age-associated changes in liver histology observed in 24-month-old zebrafish were also compared with 14-month-aged young zebrafish fed with ND (Appendix A).

### 3.4. Assessment of Fatty Liver Changes and Inflammation in Liver

As depicted in Figure 5A,D, a higher oil red O-stained area emerged in the SO supplemented group, suggesting the prolonged consumption of SO-associated with fatty liver changes. Likewise, a substantial oil red O-stained area appeared in the ND group after 24 months of supplementation, exhibiting age-associated fatty liver changes. Notably, a significant 2.7-fold (*p* < 0.001) and 2.5-fold (*p* < 0.001) reduced oil red O-stained area was detected in the OSO-supplemented group compared to the SO and ND-supplemented group, which signifies a constant consumption of OSO to prevent age-related fatty liver changes. Figure 5B–E depicts the IHC stained area representing IL-6 production in hepatic tissue following 24 months of ND, SO, and OSO supplementation. The higher IL-6 level was noticed in the SO-supplemented group, which is significantly 1.2 (*p* < 0.008) and 4.5-fold (*p* < 0.001) higher than the IL-6 production quantified in ND and OSO-supplemented groups, respectively. While compared to ND, OSO logged 3.3-fold (*p* < 0.001) low IL-6 production. The results imply the impact of prolonged consumption of OSO to diminish age-associated IL-6 production in the liver. The age-related alterations in fatty liver characteristics and IL-6 levels detected in 24-month-old zebrafish were compared with those in 14-month-aged young zebrafish that were fed with ND (Appendix A).

### 3.5. Evaluation of Hepatic Function Biomarkers

The blood analysis was performed to evaluate important hepatic function biomarkers. As depicted in Figure 6A, the maximum AST level was spotted in the SO-supplemented group; i.e., it was 1.2-fold higher than the AST level quantified in the ND group. Contrary to this, the OSO-supplemented group displayed the least amount of AST, i.e., significantly 1.4-fold (*p* = 0.002) and 1.7-fold (*p* = 0.004) lower than the AST level quantified in ND and SO-supplemented groups, respectively. Like the AST, the utmost ALT level was observed in the SO-supplemented group, i.e., 10.1% and 40.3% higher than the ND and OSO feed group (Figure 6B). Compared to the ND-supplemented group OSO, supplemented groups displayed 33.1% lower ALT level. The outcomes align well with the hepatic histology observations, validating OSO’s role in safeguarding the liver from age-related deterioration and establishing its non-toxic nature.

### 3.6. Evaluation of the Kidney Section

The results of H&E staining (Figure 7A) revealed a densely packed renal structure with a distinct distal and proximal tubule in the kidney section of ND and OSO-supplemented groups. However, compared to OSO a, luminal cell debris (indicated by the red arrow) was noticed in the tissue section of the ND-supplemented group. In contrast to ND and OSO, a loosely packed and sparsely populated renal tissue structure with certain lumen cell debris (indicated by a red arrow) was noticed in the tissue section of the SO-supplemented group. However, no sign of severe nephrotoxicity was observed in any groups, which was evidenced by the absence of proximal convoluted tissue, severe lumen cell debris, and basophilic clusters comprising new nephron formation. Results imply no adverse effect of prolonged OSO supplementation on kidney functionality.

The senescence examined by SA-β-gal activity revealed the highest senescent stained area (indicated by blue arrows) in the SO-supplemented group, which was followed by the ND and OSO-supplemented group, and it signifies the provocative role of SO consumption on the age-related senescence in the kidney (Figure 7B,E). On the contrary, a prolonged supplementation of OSO had an inhibitory effect against age-associated senescence, which was evidenced by a significantly 2.5-fold (*p* = 0.004) reduced senescence-stained area compared to the ND-supplemented group. Also, OSO supplementation displayed a significantly 2.9-fold (*p* = 0.002) lower senescence-stained area than the SO-supplemented group, signifying the antiaging effect of OSO supplementation.

The DHE fluorescent staining depicted in Figure 7C,F revealed the highest ROS production in the SO-supplemented group, which was followed by the ND-supplemented group. The least ROS production was observed in the OSO-supplemented group, which is significantly 1.3-fold (*p* = 0.041) and 1.8-fold (*p* = 0.018) less than the ROS production noticed in the ND and SO-supplemented groups, signifying the impact of OSO on diminished age-induced ROS production in the kidney.

The AO staining represents the elevated extent of apoptosis in the SO-supplemented group, which is significantly 1.1-fold (*p* < 0.001) and 1.6-fold (*p* < 0.001) higher than the extent of apoptosis in ND and OSO-supplemented groups, respectively (Figure 7D,G). While compared to the ND, the OSO supplementation exhibited a noteworthy 1.4-fold reduction in AO fluorescent intensity, testifying to the potential of OSO in mitigating age-induced apoptosis in the kidney.

### 3.7. Evaluation of the Ovary Section

The H&E staining (Figure 8A,B) showed a higher prevalence of previtellogenic in the ND and SO-supplemented groups, i.e., significantly 1.7-fold more elevated than the previtellogenic-oocytes that appeared in the OSO-supplemented group. Contrary to this, early vitellogenic oocyte presence is approximately 2.3-fold higher in the OSO-supplemented group compared to ND and OSO-supplemented groups. Likewise, the number of mature vitellogenic is approximately five-fold higher in the OSO-supplemented groups compared to the ND and SO-supplemented groups. In addition, degenerated/atretic previtellogenic (indicated by the red arrow) was noticed in the ND and SO-supplemented groups. Results described the prevention of ovary architecture against age-impaired effect in the OSO-supplemented group.

The SA-β-gal staining depicted in Figure 8A,C revealed the high prevalence of blue-stained senescence area (indicated by the red arrow) in the ovary section of the SO-supplemented group. In comparison to the SO-supplemented group, a significantly 4.3-fold (*p* < 0.05) and 6.1-fold (*p* < 0.037) lower senescence area was detected in the ND and OSO-supplemented groups, respectively. In contrast to the ND-supplemented group, the OSO-supplemented group exhibited a 1.4-fold reduction in senescence area, indicating that the consumption of OSO does not pose any toxicological implication leading to ovarian senescence. The DHE fluorescent staining revealed a significantly 1.6-fold (*p* < 0.007) and 2.6-fold (*p* < 0.001) higher ROS production in the SO-supplemented group compared to the ND and OSO-supplemented groups (Figure 8A,D). The zebrafish supplemented with OSO exhibited the lowest production of ROS, which was significantly 1.5-fold (*p* = 0.001) lower than the ND-supplemented group.

Similar to the DHE staining, an elevated extent of apoptosis was noticed in the SO-supplemented groups compared to the ND-supplemented group (Figure 8A,E). The least apoptosis was evaluated in the OSO-supplemented groups, i.e., 1.1-fold and 2.4-fold (*p* < 0.001) lower than that appeared in the ND and SO-supplemented groups.

### 3.8. Evaluation of the Testis Section

The H&E staining results revealed a loosely arranged tubular structure with inchoate/nebulous spermatocytes and spermatozoa in the testis of ND and SO-supplemented groups (Figure 9A). A higher interstitial gap between the seminiferous tubules was noticed in ND and SO-supplemented groups (Figure 9A,B,E). Compared to the ND-supplemented group, a significantly 1.4-fold higher (*p* < 0.001) interstitial space was observed in the SO-supplemented group, testifying to the adversity of persistent high SO consumption on the testis morphology. Contrary to ND and SO groups, the OSO-supplemented group displayed a compact tubular structure with well-organized/arranged spermatocytes and spermatozoa. In the OSO-supplemented group, a significantly 1.8 (*p* < 0.001) and 2.6-fold (*p* < 0.001) smaller interstitial space between the seminiferous tubules was noticed as compared to the ND and SO-supplemented groups, respectively (Figure 9B,E). The results endorsed that OSO had no adverse effect on the testis; instead, it prevented age-associated detrimental effects on the testis.

The DHE fluorescent staining revealed a significantly higher production of ROS in the ND-supplemented group that accounts for 1.3-(*p* < 0.108) and 1.8-fold (*p* < 0.01) higher ROS values than the ROS level detected in the ND and OSO-supplemented groups (Figure 9C,F). Consistent with this, an approximately 2-fold (*p* < 0.001) greater extent of apoptosis was quantified in the ND and OSO-supplemented groups compared to the SO-supplemented groups (Figure 9D,G).

### 3.9. Plasma Lipid Profile

As depicted in Figure 10, the least amount of TC was observed in the OSO-supplemented group, i.e., significantly 12% (*p* < 0.001) and 14% (*p* < 0.001) lower than the TC level quantified in the ND and SO-supplemented groups, respectively. Interestingly, no significant change in the HDL-C level was noticed in the ND, SO, and OSO-supplemented groups. Contrary to this, the HDL/TC level was 32% and 34% higher in the OSO-supplemented groups than in the ND and SO-supplemented groups. The HDL/TC values in the ND and OSO groups were nearly the same. The minimum TG level was observed in OSO, i.e., significantly 22.9% (*p* = 0.035) and 35.9% (*p* = 0.007) lower than the TG levels of the ND and SO-supplemented groups. Meanwhile, compared to SO, the ND group showed a 12% reduced level of TG. Likewise, the lowest level of LDL-C was detected in OSO, which is 12% (*p* < 0.001) and 13% (*p* < 0.001) lower than that of the ND and SO groups, respectively. A nearly similar amount of LDL-C was detected in the ND and SO groups. The TG/HDL-C level was significantly 13% (*p* < 0.001) lower in the OSO group compared to the SO group. While compared to ND and SO, a non-significant level of TG/HDL-C was observed. A combined outcome suggests no adverse effect of prolonged OSO supplementation on the lipid profile; in fact OSO was observed to exert a positive effect on blood lipid management altered by age-related stress.

## 4. Discussion

Ozonated oils are made by pumping the ozone in the oil, where the ozone reacts with the unsaturated side of the fatty acids [7], rendering the functionality of oil. Numerous studies have defined the varied therapeutic potential of ozone [26,27]. In our recent studies, we observed OSO as possessing antimicrobial [8], antioxidant, anti-inflammatory [9,10], and wound-healing activity [10]. Extending our earlier finding herein, we have evaluated the impact of prolonged (2 years) consumption of OSO on zebrafish survivability, body weight, liver, kidney, testis, ovary, and age-associated parameters to ascertain whether OSO is toxic or non-toxic when included in the diet. Results outlined the positive effect of OSO on the survivability of zebrafish; contrary to this, the prolonged consumption of SO showed a negative impact on the survivability of zebrafish. The time-dependent reduced survivability in all the groups, including the ND-supplemented groups, is due to the age, as the average age of zebrafish is merely 3 years (36 months) [28,29]. Few studies have been conducted to investigate the prolonged kinetics of zebrafish survival extensively. In one such study, zebrafish (n = 400) were tested for survivability for 45 months [28]. The findings indicate that the mean age of the tested zebrafish was 31 months (approximately 2.5 years), while only 10% of the population survived up to 41 months, and 100% of mortalities were reported at 45 months (about 3.7 years) [28]. In another study, the impact of seven different diets was tested on the survivability of zebrafish up to 9 weeks (approximately 2 months) [30]. The outcome suggests 66.7 to 100% survival over 9 weeks (about 2 months). Signifying the importance of diet on the survivability of zebrafish. So far, no study revealed the direct comparative effect of ND and OSO and SO-supplemented dietary intake on the survivability of zebrafish. However, our earlier report [9] demonstrated that OSO efficiently prevented the high-cholesterol diet (HCD)-induced mortality of zebrafish in a time-dependent manner (up to 6 months). The results align with the current findings, indicating the positive impact of OSO on zebrafish survivability whether subjected to HCD or aging-related stress. Similar to the survivability results, the consumption of OSO has an anti-obesity effect and maintains the body weight. The study unequivocally established that the existence of ozone-catalyzed compounds in OSO affects zebrafish longevity and body weight management, which are in accordance with the earlier reports highlighting the imperative role of OSO in preventing high-cholesterol diet (HCD)-induced mortality and body weight maintenance [9].

The average age of zebrafish is approximately 3 years [29], and spinal deformities are a typical age-associated feature of zebrafish [29,31] that can serve as an indirect marker of aging. The OSO consumption was found to prevent spinal curvature, thus suggesting the anti-aging effect, which strengthens the previous findings of the zebrafish survivability results. The swimming patterns of zebrafish are intricately linked to their muscle mass, which tends to decline with age [32]. Consequently, the abnormalities in muscle mass induced by aging explicitly affect swimming activity. The OSO displayed much better swimming activity than the ND and SO-supplemented groups, indirectly suggesting better muscle mass and less muscle abnormalities. The swimming activity outcome supports the spinal bending results, which are plausibly linked with muscular deformities [29,31]. A combined output of zebrafish survivability, spinal deformities, and swimming behavior strengthens the notion that the OSO can prevent/delay age-associated detrimental effects.

Furthermore, the effect of OSO consumption on the liver was evaluated. The finding concludes the non-hepatotoxic effect of OSO; even more, the OSO displayed a hepatoprotective effect against age-associated stress, which was evident by a low H&E-stained area. During age, an elevated level of ROS and a compromised antioxidant system leads to oxidative stress, which was considered the major contributor to age-associated detrimental effects on different organs [33]. We observed that OSO consumption efficiently prevents ROS generation, signifying OSO’s antioxidant effect. Mitochondria is the primary source of inheritance ROS generation and is a key contributor to age-related apoptosis [34]. Concurrently, the aging process is associated with endoplasmic reticulum (ER) stress, affecting lipid metabolism and resulting in apoptosis and the onset of fatty liver disease [35]. Therefore, an antioxidant that efficiently scavenges ROS has a healing effect against oxidative stress indued apoptosis and fatty liver changes. We observed a diminished ROS level, reduced apoptosis, and fatty liver changes in the OSO-supplemented groups, strengthening the notion that the antioxidant nature of OSO is a putative mechanism of hepatoprotection against age-induced adversity. The results aligned well with our previous findings suggesting the antioxidant nature of OSO to prevent the ROS-induced apoptotic cell death of mouse brain microglial (BV-2) cells [8]. Furthermore, the antioxidant properties of OSO are substantiated by previous investigations, demonstrating the OSO’s ability to scavenge free radicals directly [8] and prevent the zebrafish hepatic damage and fatty liver changes induced by carboxymethyllysine (CML)-induced oxidative stress [9].

Cytokines-mediated inflammation is often observed in the liver with age [34]. There is strong evidence of the age-related elevation of IL-6 [36]; in one such study, a 2.5-fold higher IL-6 production was detected in humans of 85 years and older compared to the human subjects in their 60s and 70s [37]. In the present study, we observed an age-associated elevated IL-6 level in the hepatic tissue of the zebrafish. Nonetheless, the consumption of OSO exhibited a potent anti-inflammatory effect against age-associated IL-6 production. Due to its antioxidant properties, the OSO effectively scavenges ROS, thereby reducing the production of IL-6. The notion is strongly supported by earlier studies suggesting the profound impact of oxidative stress on the induction of inflammation and inflammation-related disorders [38,39,40]. Contrary to OSO, the SO displayed a higher production of ROS, thus having a higher level of IL-6. The findings align closely with the previous findings, deciphering the OSO antioxidant properties as a critical contributor to inhibiting CML-induced IL-6 levels in the liver of zebrafish [9]. The results demonstrated that the presence of ozone-catalyzed compounds in OSO improves the biofunctionality of SO to deal with age-associated stress.

Mounting evidence proves the accumulation of senescent cells with aging; thus, cellular senescence is considered an important biomarker of aging [41]. In the present work, the least cell senescence was observed in the OSO-supplemented group; in contrast, a high prevalence of senescence was observed in the SO-supplemented group. Several intrinsic and extrinsic factors contribute to senescence, and oxidative stress is one of the leading causes of the induction of premature senescence [41]. We assert that the OSO antioxidant properties hinder the generation of ROS, thereby preventing oxidative stress and subsequently inhibiting the senescence process. The senescence in the hepatocyte is also associated with hepatic fat accumulation, cytokine secretion, and proinflammatory factors, leading to fatty liver disease and steatosis [42,43]. Therefore, inhibition in cellular senescence emerges with preventing fatty liver changes. The observed increase in proinflammatory IL-6 expression in the SO-supplemented group is correlated with significant fatty liver changes, which was possibly attributed to elevated hepatic senescence. Contrary to this, OSO diminishes hepatic senescence owing to its antioxidant nature, thus preventing fatty liver changes and the production of proinflammatory IL-6. In general, all the findings of the hepatic function confirm the hepatoprotective role of OSO by inhibiting age-associated oxidative stress that consequently diminishes the IL-6 production, fatty liver changes, and hepatocyte senescent.

In line with the hepatic tissue analysis, there were no signs of nephrotoxicity following extended OSO supplementation. Contrary to this, a minor histological change was observed in the ND-supplemented group, signifying the impact of aging on the kidney. Aging is a critical intergenic factor that elevates oxidative stress in the kidney [44,45]; consistent with the notion, we also noticed elevated oxidative stress (measured by ROS quantification) in the kidney of the ND-supplemented zebrafish. The prolonged supplementation of SO was observed to augment the age-associated ROS level. In contrast, OSO supplementation displayed a reduced ROS level, which was significantly better than the ND-supplemented group, signifying the importance of ozone-catalyzed compounds to maintain age-altered oxidative stress in the kidney. Also, aging significantly contributes to apoptosis in the kidney via caspase 3 activation, leading to kidney function impairment [44,46]. Therefore, a substance that halts excessive apoptosis in the kidney improves kidney function. The OSO showed a much lower extent of apoptosis than the ND group, suggesting the positive impact of OSO supplementation on preserving kidney functionality against age. As oxidative stress is the primary culprit in inducing the apoptosis and cell senescence following different detrimental effects [47], the antioxidant nature of OSO, evidenced by present findings and documented in the previous reports [8,9], seems to be a key contributor against age-induced apoptosis and cellular senescence in the kidney.

Oxidative stress, inflammation, and apoptosis typically manifest in the aged testis [48], and these factors substantially influence both the quality and quantity of sperm along with the process of spermatogenesis [48]. Therefore, the inhibition of apoptosis and oxidative stress in testis leads to preventive events against age-associated impairment. OSO-guided ROS and apoptosis inhibition is the major reason for the age-associated deterioration of testis.

Increasing age significantly affects the functionality and metabolic disorders in ovaries [49]. In the aged ovaries, a high prevalence of sphingomyelin is observed, which leads to critical events in apoptosis [49]. Also, a decreased antioxidant status (reduced glutathione peroxidase and superoxide dismutase levels) was noticed in the aged ovaries that are believed to elicit oxidative stress [49], leading to ovarian dysfunction [50]. The high level of oxidative stress can lead to telomere damage and mitochondrial dysfunction [50] that can lead to cellular senescence in the aged ovaries [51,52]. Therefore, a substance that can counter oxidative stress, apoptosis, and inflammation can induce preventive cellular events against age-associated ovarian damage. We posit that OSO, by virtue of its antioxidant and anti-inflammatory functions, safeguards the ovaries against the adverse impacts of aging. These findings aligned well with the previous reports documenting the impact of varied substances to prevent ovary functionality due to their antioxidant [49,50,53] and anti-inflammatory role [52,53].

Accumulating literature suggested the adverse effect of aging on the blood lipid profile [54] that significantly contributes to the onset of cardiovascular diseases [54]. In general, the enhancement of blood TC and LDL-C with the reduction in the HDL-C level are the prominent features of old age [55]. The exact mechanism of age-allied hypercholesteremia has yet to be well understood. However, among the several reasons fatty acids induced insulin resistance, growth hormone deficiency, a decrease in peroxisome proliferator-associated receptor α, and liver dysfunction have a notable impact on age-related dyslipidemia [55]. The importance of proinflammatory cytokines like IL-6 has been documented to alter the blood lipid profile [56], signifying the importance of inflammation in dyslipidemia. In the present work, we have observed the reduced level of serum TC, TG, and HDL-C in the OSO-supplemented group along with an alleviated level of IL-6 in the hepatic tissue, which might be a reason for the low lipid level in the OSO-supplemented zebrafish. This notion is firmly substantiated by previous reports revealing a distinct link between inflammatory disorders and the blood lipid profile [57]. We postulated that the anti-inflammatory effect exerted by OSO (suggested by IHC staining) is a pivotal event in maintaining the blood lipid profile in the OSO supplementation zebrafish.

## 5. Conclusions

A two-year OSO consumption displays a non-toxic effect against zebrafish survivability and functionality of vital organs. Moreover, OSO consumption was logged for the preventive effect against age-associated spinal deformities, senescence, and the function of vital organs that found significantly better than the consumption of SO, suggesting the importance of ozone-catalyzed compounds to improve the functionality of SO. The antioxidant and anti-inflammatory nature of OSO was a key regulator responsible for the beneficial effects against age-associated adversity of spinal bending, swimming activity, and dyslipidemia (Figure 11). The results endorse OSO’s safe and non-toxic nature and propose it as a nutraceutical to prevent age-associated deterioration.

## Figures and Tables

**Figure 1 antioxidants-13-00123-f001:**
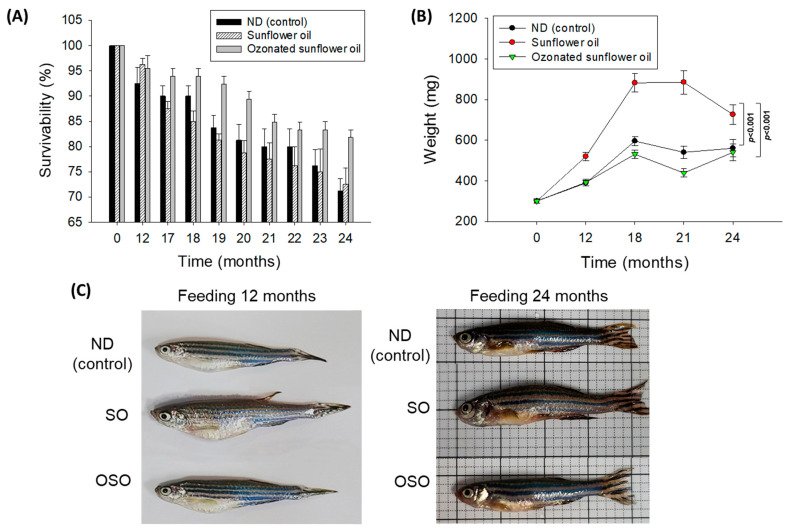
Long-term supplementation of sunflower oil (SO) and ozonated sunflower oil (OSO) on the survivability and body weight of zebrafish. (**A**) Time-dependent survivability. Data were expressed as mean ± SEM. (**B**) A time-dependent change in body weight. (**C**) Morphology of zebrafish at 12 and 24 months feeding. Data were expressed as mean ± SEM. ND represents the control normal diet; SO represents ND supplemented with 20% SO (*wt*/*wt*); OSO represents ND supplemented with 20% OSO (*wt*/*wt*). The *p* value signifies the statistical significance discerned between groups (based on the diet and time of feeding) resulting from the two-way ANOVA following Tukey’s post hoc analysis.

**Figure 2 antioxidants-13-00123-f002:**
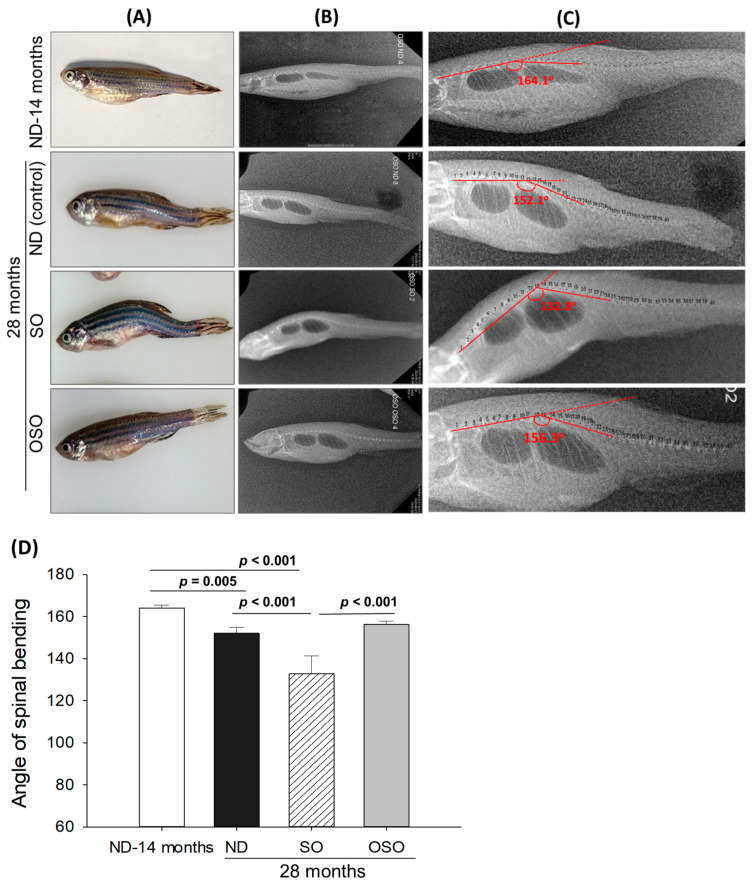
Age-related structural deformities in zebrafish after 28 months of supplementation of sunflower oil (SO) and ozonated sunflower oil (OSO). (**A**) A pictorial view representing structural deformities. (**B**) X-ray images and (**C**) and digitally zoomed X-ray images of the zebrafish skeletal (backbone). (**D**) The angle of the spinal deformities (bending) was determined employing image J software version 1.53 (http://rsb.info.nih.gov/ij/, accessed on 16 June 2023). ND represents the control normal diet; SO represents ND supplemented with 20% SO (*wt*/*wt*); OSO represents ND supplemented with 20% OSO (*wt*/*wt*). The *p* value signifies the statistical significance discerned between groups resulting from the one-way ANOVA following Tukey’s post hoc analysis.

**Figure 3 antioxidants-13-00123-f003:**
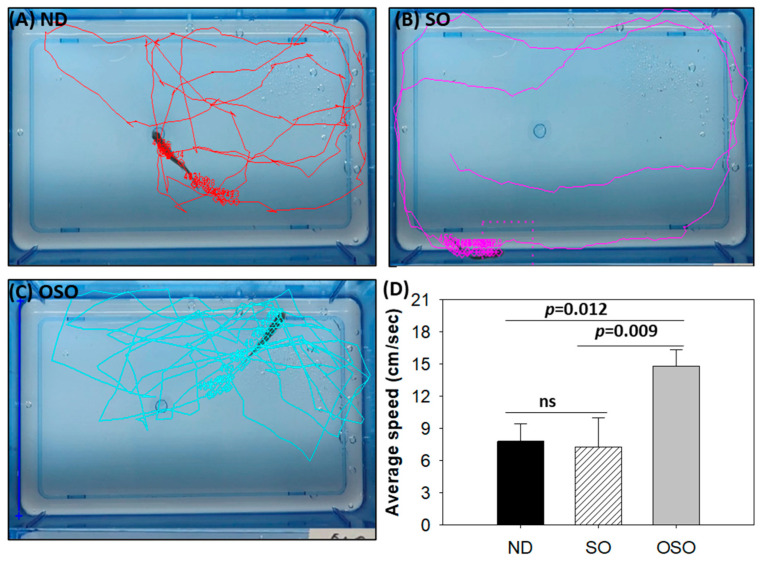
Swimming trajectory and swimming speed of zebrafish after 28 months of supplementation of (**A**) normal diet (ND), (**B**) normal diet + 20% SO (*wt*/*wt*) and (**C**) normal diet + 20% OSO (*wt*/*wt*). (**D**) Average swimming speed of the zebrafish. Tracker video analysis and modeling tool version number 6.1.5 (available at https://physlets.org/tracker accessed on 16 May 2023) was utilized to determine the swimming trajectory and swimming speed of zebrafish. ND represents the control normal diet; SO represents ND supplemented with 20% SO (*wt*/*wt*); OSO represents ND supplemented with 20% OSO (*wt*/*wt*). The *p*-value signifies the statistical significance discerned between groups resulting from the one-way ANOVA following Tukey’s post hoc analysis; ns represent non-significant differences between the groups.

**Figure 4 antioxidants-13-00123-f004:**
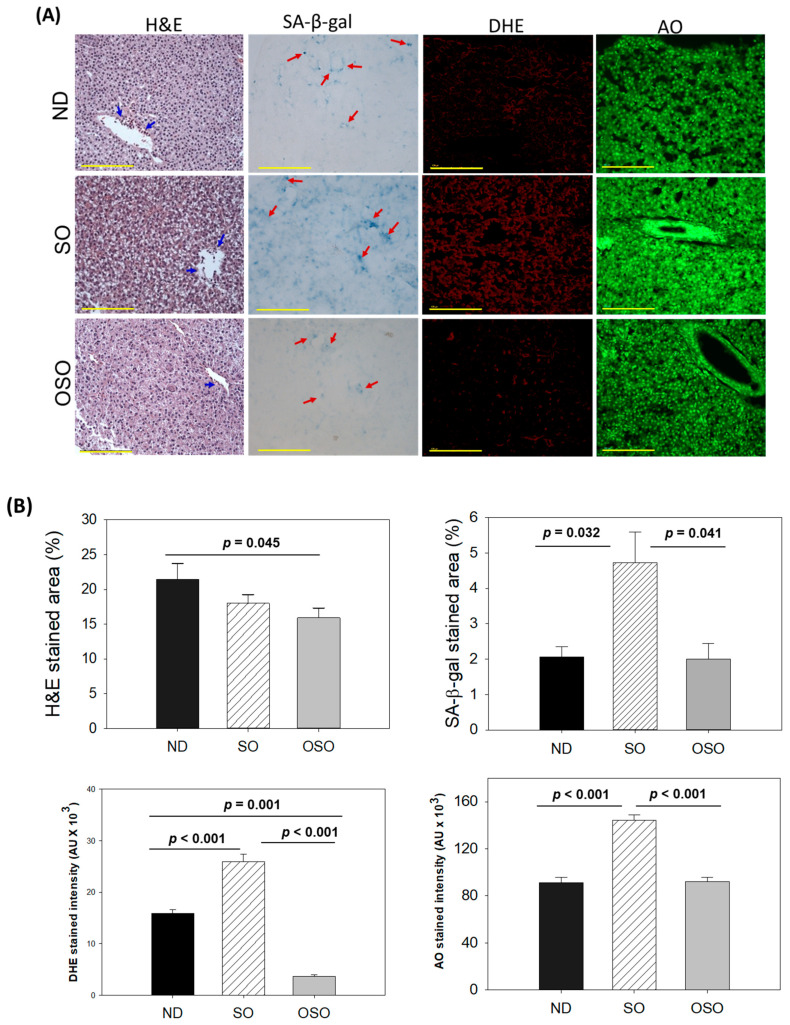
A comparative assessment of 24 months of supplementation of sunflower oil (SO) and ozonated sunflower oil (OSO) on the hepatic tissue of zebrafish. (**A**) Hematoxylin and eosin (H&E) staining, senescence-associated β galactosidase (SA-β-gal) staining, dihydroethidium (DHE) and acridine orange (AO) fluorescent staining (100 μm, yellow scale bar). Red arrow indicates SA-β-gal positive cells. Scale bar indicates 100 μm. (**B**) Image J-based quantification of H&E and SA-β-gal-stained area, fluorescent intensities of DHE and AO-stained area. ND represents the control normal diet; SO represents ND supplemented with 20% SO (*wt*/*wt*); OSO represents ND supplemented with 20% OSO (*wt*/*wt*). The *p* value signifies the statistical significance discerned between groups resulting from the one-way ANOVA following Tukey’s post hoc analysis.

**Figure 5 antioxidants-13-00123-f005:**
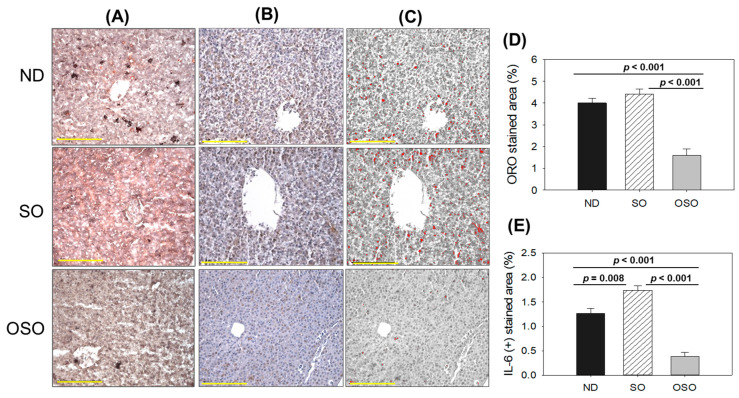
A comparative effect of 24 months of supplementation of sunflower oil (SO) and ozonated sunflower oil (OSO) on fatty liver alterations and IL-6 production in zebrafish. Yellow scale bar indicates 100 μm. (**A**) Oil red O staining. (**B**) Immunohistochemistry (IHC) for the assessment of IL-6 generation. (**C**) Employing Image J software, the brown color of the native IL-6-stained area has interchanged with red color, employing a brown color threshold value from 20 to 120 to intensify the visualization of the IHC stained area. (**D**,**E**) Quantification Oil red O stained and IL-6-stained area employing Image J software. ND represents the control normal diet; SO represents ND supplemented with 20% SO (*wt*/*wt*); OSO represents ND supplemented with 20% OSO (*wt*/*wt*). The *p* value signifies the statistical significance discerned between groups resulting from the one-way ANOVA following Tukey’s post hoc analysis.

**Figure 6 antioxidants-13-00123-f006:**
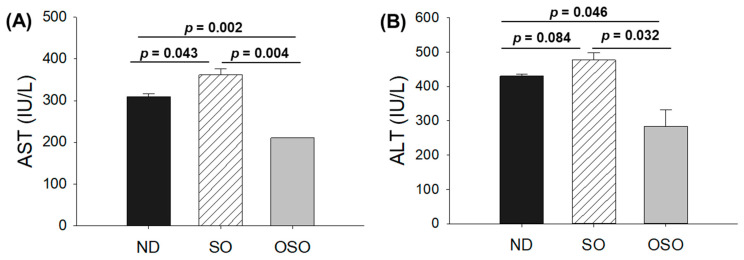
A comparative effect of 24 months of supplementation of sunflower oil (SO) and ozonated sunflower oil (OSO) on the hepatic function biomarkers: (**A**) aspartate aminotransferase (AST) and (**B**) alanine aminotransferase (ALT). ND represents the control normal diet; SO represents ND supplemented with 20% SO (*wt*/*wt*); OSO represents ND supplemented with 20% OSO (*wt*/*wt*). The *p* value signifies the statistical significance discerned between groups resulting from the one-way ANOVA following Tukey’s post hoc analysis. The *p* value signifies the statistical significance discerned between groups resulting from the one-way ANOVA following Tukey’s post hoc analysis.

**Figure 7 antioxidants-13-00123-f007:**
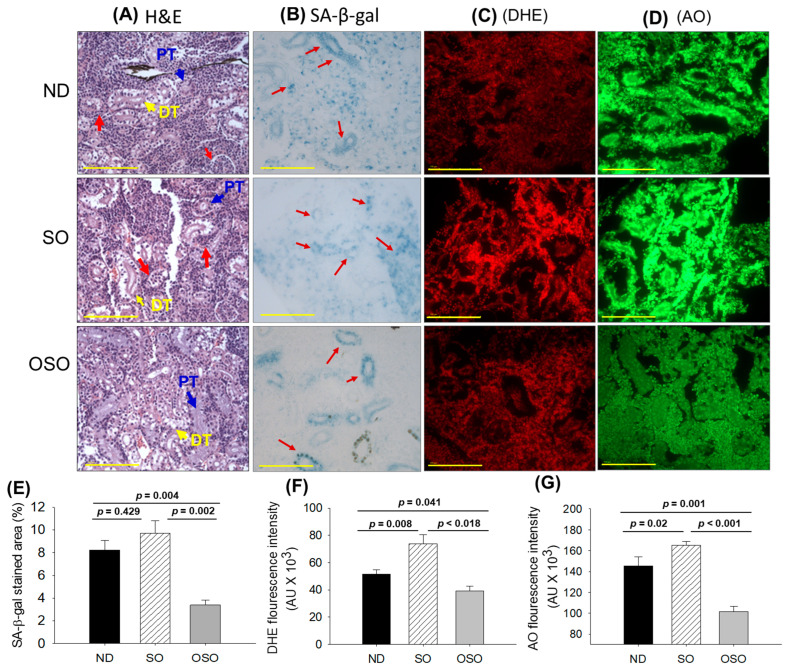
A comparative effect of 24 months supplementation of sunflower oil (SO) and ozonated sunflower oil (OSO) on the kidney of zebrafish. Yellow scale bar indicates 100 μm. (**A**) Hematoxylin and eosin (H&E) staining. PT and DT symbolize the proximal and distal tubules, respectively. The red arrows indicate luminal debris. (**B**) Senescence-associated β galactosidase (SA-β-gal) staining. Red arrows indicate senescent area. (**C**) Dihydroethidium (DHE) fluorescent staining. (**D**) Acridine orange (AO) fluorescent staining. Quantification of (**E**) SA-β-gal-stained area, (**F**) DHE and (**G**) AO-stained area utilizing Image J software. ND represents the control normal diet; SO represents ND supplemented with 20% SO (*wt*/*wt*); OSO represents ND supplemented with 20% OSO (*wt*/*wt*). The *p* value signifies the statistical significance discerned between groups resulting from the one-way ANOVA following Tukey’s post hoc analysis.

**Figure 8 antioxidants-13-00123-f008:**
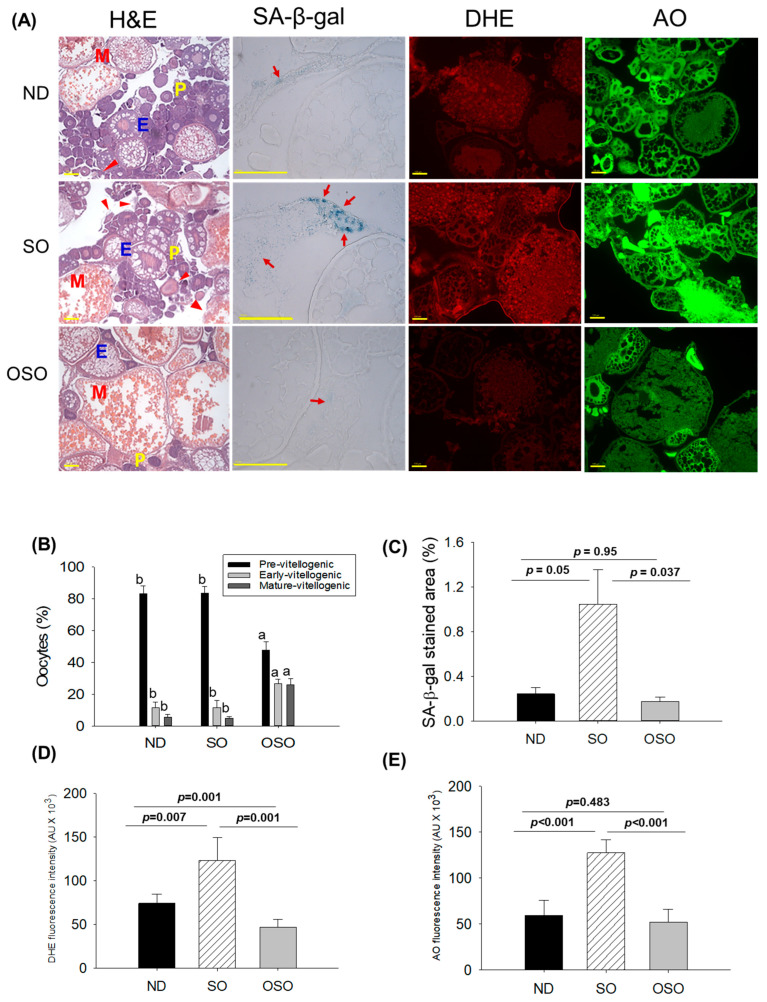
A comparative effect of 24 months of supplementation of sunflower oil (SO) and ozonated sunflower oil (OSO) on the ovary of zebrafish. (**A**) Ovarian cell morphology was analyzed by hematoxylin and eosin (H&E) staining (P, E, and M represents pre, early and mature vitellogenin stages, respectively), senescence-associated β galactosidase (SA-β-gal) staining (red arrow indicating senescent area.), dihydroethidium (DHE) and acridine orange (AO) fluorescent staining. (100 μm, yellow scale bar). (**B**) Quantification of the different developmental stages of oocytes based on the H&E images. (**C**–**E**) Image J software-based quantification of SA-β-gal-stained area, DHE and AO-stained area, respectively. ND represents the control normal diet; SO represents ND supplemented with 20% SO (*wt*/*wt*); OSO represents ND supplemented with 20% OSO (*wt*/*wt*). The alphabets (a,b) above the bar graphs showed the statistical difference between the groups. The *p* value signifies the statistical significance discerned between groups resulting from the one-way ANOVA following Tukey’s post hoc analysis.

**Figure 9 antioxidants-13-00123-f009:**
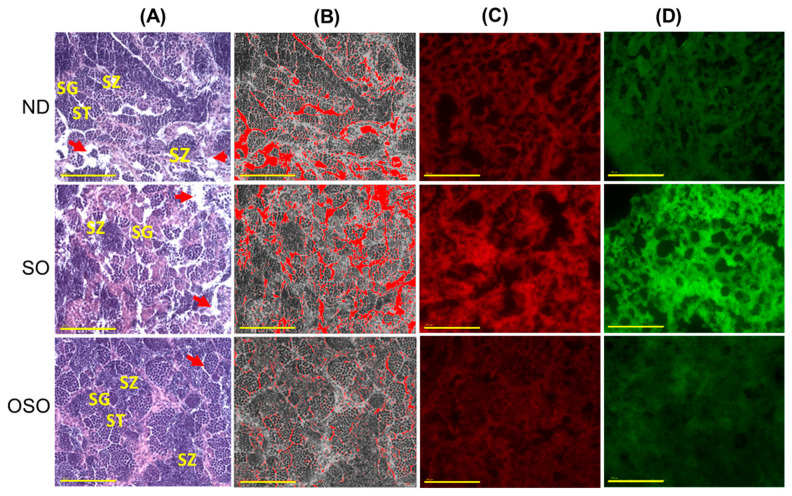
A comparative effect of 24 months of supplementation of sunflower oil (SO) and ozonated sunflower oil (OSO) on the zebrafish testis. Yellow scale bar indicates 100 μm. (**A**) Testis histology was analyzed by hematoxylin and eosin (H&E) staining. SG, ST, and SZ represent spermatogonia, spermatocytes and spermatozoa. The red arrow specifies the interstitial space between seminiferous tubules. (**B**) Transformation of the white color (void space) that appeared in the H&E section to the red color for the clear visualization of interstitial space between seminiferous tubules (at the threshold value of 220–255) employing Image J software. (**C**,**D**) DHE and AO-stained area, respectively. (**E**) Image J-based quantification of interstitial space in testis. (**F**,**G**) Quantification of dihydroethidium (DHE) and acridine orange (AO) fluorescent stained area, respectively, utilizing Image J software. ND represents the control normal diet; SO represents ND supplemented with 20% SO (*wt*/*wt*); OSO represents ND supplemented with 20% OSO (*wt*/*wt*). The *p* value signifies the statistical significance discerned between groups resulting from the one-way ANOVA following Tukey’s post hoc analysis.

**Figure 10 antioxidants-13-00123-f010:**
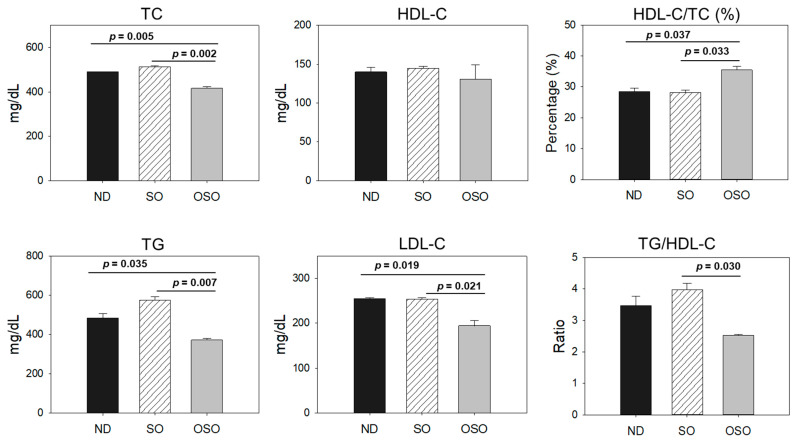
Blood lipid profile of zebrafish fed with sunflower oil (SO) and ozonated sunflower oil (OSO) for 24 months. TC (total cholesterol), TG (triglycerides), HDL-C (high-density lipoprotein cholesterol) and LDL-C (low-density lipoprotein cholesterol), ND represents the control normal diet; SO represents ND supplemented with 20% SO (*wt*/*wt*); OSO represents ND supplemented with 20% OSO (*wt*/*wt*). The *p* value signifies the statistical significance discerned between groups resulting from the one-way ANOVA following Tukey’s post hoc analysis.

**Figure 11 antioxidants-13-00123-f011:**
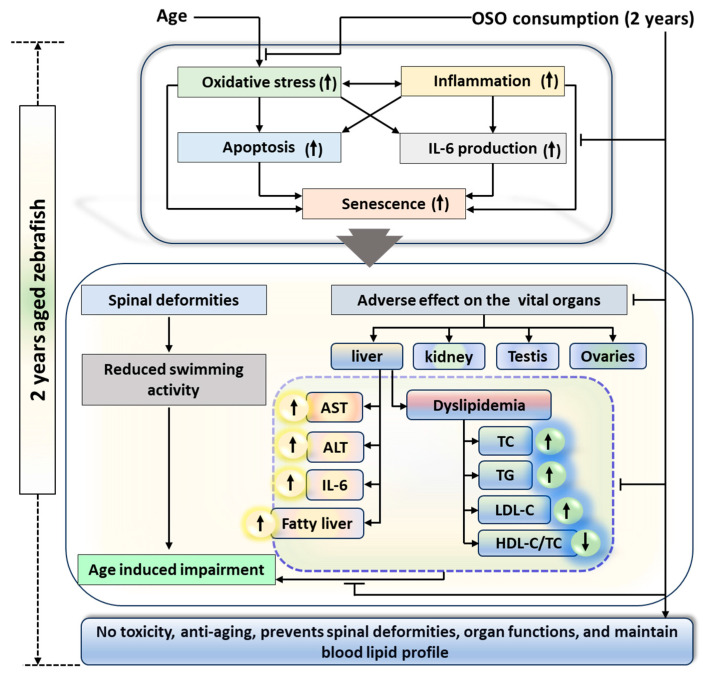
Impact of ozonated sunflower oil (OSO) consumption for 2 years on the aging and senescence associated changes in zebrafish. AST, aspartate aminotransferase; ALT, alanine aminotransferase; IL-6, interleukin-6; HDL-C, high-density lipoprotein-cholesterol; LDL-C, low-density lipoprotein-cholesterol TC, total cholesterol; TG, triglyceride.

## Data Availability

The data used to support the findings of this study are available from the corresponding author upon reasonable request.

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
