# Peer review of "Prolonged Supplementation of Ozonated Sunflower Oil Bestows an Antiaging Effect, Improves Blood Lipid Profile and Spinal Deformities, and Protects Vital Organs of Zebrafish (Danio rerio) against Age-Related Degeneration: Two-Years Consumption Study"

_antioxidants, 2024, doi:10.3390/antiox13010123_

Round 1

Reviewer 1 Report

Comments and Suggestions for Authors

This submission reports the antiaging  effects of ozonated sunflower oil on zebrafish.  The overall quality can be improved by adding some quantitative analysis and therefore needs a major revision before acceptance. The points that need revisions are:

1.  Why the survivability of control zebrafish decreased so rapidly after 19 months? Did the authors compare their results with those of other investigators?  As no standard deviation is shown  for each time point under each condition, it would be much better to replace figure 1A with a bar graph and includes the survivability of control zebrafish reported by other researchers as a comparison. 

2. Why zebrafish fed with SO-supplemented diet look so different? More zebrafish under each condition should be shown in figure 1C to convince the readers. 

3. It would be better to add a figure  that illustrates the relationship between aging of control zebrafish and the presence of ROS and apoptosis or lipid biomarkers for readers to understand what damages are caused by aging alone.  

Comments on the Quality of English Language

The quality of English is fine.

Author Response

Thank you for your valuable comments and suggestions. 

Please find attached doc as point-to-point response.

Reviewer 2 Report

Comments and Suggestions for Authors

The authors present a two-years work of prolonged supplementation of ozonated sunflower oil on adult zebrafish. During the experimental period, physiological and morphological parameters were evaluated. After experiment, histological examination of different organs as well as histochemical analysis of different biomarkers was conducted. Overall, the results show no adverse effects on zebrafish health while some protective effects were observed.

Although presenting novel and interesting findings, some issued need to be solved:

-          L52, what is the relation with the objective of the work? What was observed in the study cited?

-          L100, why were these concentrations/percentages used here?

-          L102, how was the distribution of SO and OSO guaranteed? Was any chemical analysis conducted on the diet?

-          L113, please clarify the sample size. Were these 80 animals from each group exposed to the same solution? Apparently, it seems that this was the case meaning that authors are working with a n=1 instead of a minimum o n=3.

-          L119, how was the weight recorded and what was the interval between measurements? What is the impact of 2-phenoxyethanol on these repeated experiments and on zebrafish welfare? Were the same animals used or were them picked randomly?

-          L150, please include further details on the antibodies used (e.g. dilution, brand, etc).

-          L165, the cited study used live embryos rather than fixed tissues. Is the methodology similar? In addition, how may animals were analyzed?

-          L169, review the number of animals used for each experiment. Only for histology is referred a n=10.

-          L184, include further details on these kits (e.g. probes used, wavelength of measurement, etc).

-          L192, was normality checked before applying parametric tests?

-          L201-216, were these results statistically different? This section is too extensive and data does not include standard deviation or other dispersion indicator.

-          L226, was a two-way anova conducted to verify statistical differences between time-points?

-          L231, “which severely impacts body weight enhancement” is this a negative outcome?

-          L236, which morphological changes were observed?

-          L242, why was data compared to 14 months young zebrafish?

-          L258, no supplementary file was uploaded with the manuscript. In addition, how was swimming behaviour assessed?

-          L573, how can this be related with adult outcomes?

-          L576-L581, what is the relation with the aim of the work?

-          L618, why was 14-month animals tested for curvature and not for oxidative stress-related parameters? This would be important to verify the effect of aging.

Author Response

(The authors gave the same response as above.)

Round 2

Reviewer 1 Report

Comments and Suggestions for Authors

I recommend the acceptance of this article.

Author Response

Thank you for your valuable comments and suggestions during revision. 

Thank you very much for acceptance. 

Reviewer 2 Report

Comments and Suggestions for Authors

The authors have provided a point-by-point reply to my comments and have improved the manuscript based on the reviewers’ comments. However, I still have some doubts regarding the manuscript:

-          The rationale for using the concentrations does not convince me as the screening investigation, which the authors mention as “data not shown”, should be available and shown to verify that up to 20% of SO and OSO can be adsorbed in the diet. This is in line with the following question regarding the distribution of SO and OSO. Without further chemical analysis of the diet, there is no guarantee of the percentage that is adsorbed to the diet. Without this quantification, the results are not meaningful.

-          As requested before, error bars must be included in Figure 1.

-          As referred before, there are two variables in study: group/treatment and time. As such, a two-way ANOVA should be implemented.

-          Swimming behavior should be quantified and not qualitatively assessed.

Author Response

Thank you for your valuable comments and suggestions. 

Please find attached doc as point-to-point response for 2nd revision.

Round 3

Reviewer 2 Report

Comments and Suggestions for Authors

While the authors have replied satisfactorily to most of my comments and have included further data to increase the validity of this study, I still have some concerns regarding behavioural data. In fact, these results should be included in the main text and statistics done. Furthermore, the variation assotiated with the value seems a low based on the number of animals tested (n=20 for each group in 3 replicates). As such, this data should be rechecked.

Author Response

Thank you very much for your valuable comments and suggestions. 

Please find attached doc as point-to-point response as 3rd revision.

Thank you again for recognizing our efforts and your valuable suggestion to enhance the quality of the article. 

Round 4

Reviewer 2 Report

Comments and Suggestions for Authors

The authors have provided new data and have reformulated the manuscript according to my comments. I do not have more comments on this work.